# Anticancer Activity of Water-Soluble Olsalazine-PAMAM-Dendrimer-Salicylic Acid-Conjugates

**DOI:** 10.3390/biom9080360

**Published:** 2019-08-13

**Authors:** Sandra Cortez-Maya, Luis Daniel Pedro-Hernández, Elena Martínez-Klimova, Teresa Ramírez-Ápan, Marcos Martínez-García

**Affiliations:** 1Instituto de Química, Universidad Nacional Autónoma de México, Ciudad Universitaria, Circuito Exterior, Coyoacán, C.P. 04510, México D.F., México; 2Facultad de Química, Universidad Nacional Autónoma de México, Ciudad Universitaria, Circuito Interior, Coyoacán, C.P. 04510, México D.F., México

**Keywords:** olsalazine, salicylic acid, dendrimers, anticanceractivity

## Abstract

Improving the activity and selectivity profile of anticancer agents will require designing drug carrier systems that employ soluble macromolecules. Olsalazine-PAMAM-dendrimer-salicylic acid-conjugates with dendritic arms of different lengths have shown good stability regarding the chemical link between drug and spacer. In this study, the drug release was followed in vitro by ultraviolet (UV) studies. Evaluation of the cytotoxicity of the olsalazine-PAMAM-dendrimer-salicylic acid-conjugates employing a sulforhodamine B (SRB) assay in PC-3 (human prostatic adenocarcinoma) and MCF-7 (human mammary adenocarcinoma) cell lines demonstrated that conjugate 9 was more active as an antiproliferative agent than cisplatin, and no cytotoxicity towards the African green monkey kidney fibroblast (COS-7) cell line was observed in any of the conjugates synthesized in the present work.

## 1. Introduction

Olsalazine (3,3’-azobis salicylic acid) is an anti-inflammatory drug used in the treatment of inflammatory bowel disease (IBD), ulcerative colitis (UC) and other gastrointestinal problems [1,2]. Olsalazine consists of two 5-aminosalicylic acid (5-ASA) molecules joined by an azo bridge [2]. Olsalazine has been shown to inhibit the development of colorectal cancer in patients [2,3], and it has also been proposed as a broad-spectrum anticancer agent [4]. Colonic bacteria were found to split the azo bond of this compound through the liberation of the two molecules of 5-ASA [5,6,7,8,9]. Nanocarriers based on generation 5-poly(amidoamine) (PAMAM) dendrimers with salicylic acid covalently attached to their surface were used as chemical exchange saturation transfer (CEST) imaging agents, making them an effective non-toxic ligand that could be useful in the design of new materials such as dendrimers [10]. Dendrimers are well-defined, globular, highly branched macromolecules with a large number of functional groups at their surface. Moreover, their size, molecular weight and surface properties can be easily controlled. Dendrimers have attracted much attention in drug delivery due to their unique structural properties and good water-solubility [11,12,13,14]. Thus, 5-amino salicylic acid has been conjugated to poly(ethylene glycol) dendrimers and polyamidoamine (PAMAM) dendrimers to reduce their toxicity and improve their activity [11,12,13,14,15,16,17,18,19,20,21].

This article reports the rational design of olsalazine as a nucleus of PAMAM-dendrimer-salicylic acid-conjugates to reach human cancer cells and the evaluation of drug release from these macromolecular products. In this study, the olsalazine-PAMAM-dendrimers of second generation (G2) with 8 amino terminal groups and salicylic acid in the periphery were synthesized (Figure 1). The release of the drug from the conjugates in vitro was also determined in comparison with free olsalazine.

## 2. Materials and Methods

All reagents were purchased from Sigma-Aldrich (St. Louis, MO, USA). Salicylic acid (SA), 5-amino-2-hydroxybenzoic acid (98%), hydrochloric acid (36.5–38%), sodium nitrite (98%), hexamethylenediamine (98%), ethylendiamine titanium(IV) isopropoxide (97%), methyl bromoacetate, methyl acrylate, Triton X-100 solution and phosphate buffered saline pH 7.4 were used without additional purification. Ethanol and methanol were used without additional drying. Tetrahydrofuran (THF) was dried with Na. *N,N*-Dimethylformamide (DMF) and dimethylsulfoxide (DMSO) (99.8%) were used without any purification.

### 2.1. Instrumentation

^1^H and ^13^C NMR spectra were recorded on a Varian Unity-300 MHz with tetramethylsilane (TMS) as an internal reference. Infrared (IR) spectra were measured on a Nicolet FT-SSX spectrophotometer (Bruker, Billerica, MA, USA). Elemental analysis was determined by Galbraith Laboratories, INC Knoxville (Sycamore, MO, USA). FAB+ mass spectra were taken on a JEOL JMS AX505 HA instrument (JEOL manufacturers, Tokyo, Japan). Electrospray mass spectra were taken on a Bruker Daltonic, Esquire 6000 (Bruker, Billerica, MA, USA). Matrix-assisted laser desorption/ionization-time of flight (MALDI-TOF) mass spectra were taken on a Bruker Omni FLEX using 9-nitroanthracene (9NA) as a matrix. The ultraviolet–visible (UV-Vis) absorption spectra were obtained at room temperature with a Shimadzu 2401 PC spectrophotometer (Shimadzu, Kyoto, Japan).

### 2.2. Anticancer Screening

Human glioblastoma, U-251, PC-3 (human prostatic adenocarcinoma), K-562 (human chronic myelogenous leukemia cells), HCT-15 (human colorectal adenocarcinoma), MCF-7 (human mammary adenocarcinoma) and SKLU-1 (human lung adenocarcinoma) were supplied by the National Cancer Institute (USA). The African green monkey kidney fibroblast (COS-7) cell line was acquired from Thermo Fisher Scientific (Waltham, MA USA). Cytotoxicity assays were determined using the protein-binding dye sulforhodamine B (SRB) in microculture to measure cell growth, as described previously [19,20]. Olsalazine-PAMAM-dendrimer-salicylic acid-conjugates were prepared in fresh culture medium with 2% DMSO immediately before use, then added to the culture medium. Control cells were treated with 2% DMSO. Mean values of half maximal inhibitory concentration (IC_50_) for each tested drug were calculated from at least three experiments. 

### 2.3. In Vitro Cellular Uptake of the Conjugates of Salicylic Acid

At 35 °C in a Synergy HT Microplate Reader (Bio-Tek Instruments, Winossky, VT, USA), the cellular uptake of olsalazine-PAMAM-dendrimer-salicylic acid-conjugates was quantitatively determined in a pH 7.4 buffer solution. The human mammary adenocarcinoma cancer cells, MFC-7 were seeded in a 96-well black plate at 10,000 cells/well and cultured. After 24 h, MFC-7 cancer cells were incubated with either olsalazine-PAMAM-dendrimer-salicylic acid-conjugates or free olsalazine 1 at a concentration of either 10 or 25 μM for 16 h at 35 °C. After incubation, the suspension was removed and the cells were washed four times with 50 μL cold phosphate buffered saline solution (PBS, pH 7.4). All the resulting solutions were kept at 37 °C for 24 h prior to measurements. The fluorescence intensity of each sample well was measured by a microplate reader calibrated with standard solutions of olsalazine (0.1–50 μM/mL) under similar conditions. The excitation and emission wavelengths were 300 and 400 nm; and 340 and 500 nm, respectively. All release measurements were carried out in triplicate and the average values were plotted.

### 2.4. Hydrolytic Release of the Drug from Olsalazine-PAMAM-Dendrimer-Salicylic Acid-Conjugates

The hydrolytic release of the drugs was measured at 37 °C using 25 mL PBS, at 37 °C and pH 7.4. The olsalazine-PAMAM-dendrimer-salicylic acid-conjugates were dried in vacuum at room temperature. Figure 2 shows the time course of UV spectral change of the conjugates incubated in phosphate buffer. The maximum UV absorption of the Schiff base in PBS was at 300 nm at pH 7.4. The dendrimers and conjugates were dispersed in buffer solution and the hydrolysis was made. The mixture solution was stirred by a magnetic stirrer continuously at 600 rev/min, and a sample solution (3 mL) was withdrawn at selected intervals of time and replaced with the same volumes of media. The quantity of the released drug was analyzed by means of UV spectroscopy and determined from the calibration curve using a 1-cm quartz cell.

## 3. Synthesis of Olsalazine-PAMAM-Dendrimers

Olsalazine was obtained from salicylic acid and 5-amino-2-hydroxybenzoic acid (5-ASA) with sodium nitrite in HCl at 0 °C. The olsalazine derivative 2 was obtained using hexamethylenediamine in presence of molecular sieves of 4 Å and titanium (IV) isopropoxide in THF at 70 °C (Scheme 1).

The olsalazine-dendrimers intermediates 3 and 5 were obtained using methyl acrylate in a mixture of methanol, toluene 5:1 and reflux (Scheme 2). The olsalazine-PAMAM-dendrimers 4 and 6 with the NH_2_-terminated groups were obtained with ethylenediamine in a mixture of methanol, toluene 5:1 and reflux (Scheme 2).

The compounds 4 and 6 were characterized by ^1^H NMR spectroscopy. In the spectrum, the broad signals due to the internal -CH_2_ of the hexylenediamine groups were observed from δ_H_ 1.37 to 1.63, and the signals due to the CH_2_- groups were observed at the dendritic arms from δ_H_ 1.94 to 3.31. One triplet due to the -CH_2_ at 3.62 was also observed (see NMR spectra for all the compounds can be found in Appendix A).

### 3.1. Synthesis of Olsalazine-PAMAM-Dendrimer-Salicylic Acid-Conjugates

After that, the methyl salicylate was obtained from methyl salicylic acid in methanol and sulfuric acid at the reflux. The olsalazine-PAMAM-dendrimer-salicylic acid-conjugates 8 and 9 with methyl salicylate were obtained in a mixture of methanol/toluene at the reflux (Figure 1).

The conjugates were soluble in water and characterized by proton nuclear magnetic resonance (^1^H NMR) spectroscopy (Figure 2). The following signals were observed: from δ_H_ 1.03 to 1.86, three broad signals were observed due to the internal CH_2_- groups at the hexyl moiety; from δ_H_ 2.54 to 3.83, the signals for the CH_2_ groups assigned to the dendritic branches and the terminal CH_2_ groups at the hexyl fragment were observed; and from δ_H_ 6.5 to 7.5 the protons for the aromatic fragments were observed.

### 3.2. Hydrolytic Release of Olsalazine-PAMAM-Dendrimer-Salicylic Acid-Conjugates 8 and 9

The stability of the conjugates was studied in conditions of gastric fluid pH 1.2 and intestinal fluid pH 7.4 for 2 and 12 h, respectively, where a negligible amount of 5-ASA was hydrolyzed. No degradation of conjugates at pH 1.2 and 7.4 showed that the conjugates were stable and remained intact at the gastric pH (see Appendix A).

At pH 7.4, the olsalazine-PAMAM-dendrimer-salicylic acid-conjugates 8 and 9 showed a relatively slow release in the first minutes, followed by a sustained release after 500 min. Salicylic acid was released from the corresponding first generation 8 and the second generation 9**.**
Figure 3 shows the time course of UV spectral change from the conjugates. The amount of drug released from the conjugates was calculated from their UV absorption at 300 nm at pH 7.4. The amount of drug released was constant regardless of the generation number. This result could be related to the relative stability of the conjugates at the pH 7.4.

Drug release from both olsalazine-PAMAM-dendrimer-salicylic acid-conjugates 8 and 9 was calculated from their UV absorption at 300 nm, as depicted in Figure 4. At physiological pH 7.4, conjugate 8 showed a less controlled release pattern compared to conjugate 9.

### 3.3. Cytotoxicity of Olsalazine-PAMAM-Dendrimer-Salicylic Acid-Conjugates

The cytotoxic activity of the synthesized olsalazine-PAMAM-dendrimers 4 and 6 and the olsalazine-PAMAM-dendrimer-salicylic acid-conjugates 8 and 9 were chosen for evaluation of their biological activity against cancer cell lines. We screened in vitro against six human cancer cell lines: U251 (human glioblastoma), PC-3 (human prostatic adenocarcinoma), K-562 (human chronic myelogenous leukemia cells), HCT-15 (human colorectal adenocarcinoma), MCF-7 (human mammary adenocarcinoma) and SKLU-1 (human lung adenocarcinoma). As a control, we also tested against the COS-7 African green monkey kidney fibroblast cell line. Free olsalazine and cisplatin were used to compare the antiproliferative activity of the dendrimers. The dendrimers 4 and 6, and the conjugates 8 and 9 were not active against the cell lines U251 and SKLU-1. The compounds 4–9 showed almost the same activity against cell lines PC-3, K562, HCT-15 and MFC-7 at 25 and 10 μM. The concentration of the conjugates 8 and 9 was diluted four and eight times, respectively, to have the anticancer activity of one molecule of salicylic acid to compare with the free salicylic acid, olsalazine or cisplatin. Table 1 shows the normalized percentage of inhibition of the growth that allows comparing the activity of the same amount of salicylic acid in its free state versus when it is contained in the conjugates 8 and 9. Cisplatin was tested at a concentration of 10 µM. For the olsalazine-PAMAM-dendrimer-salicylic acid-conjugate 8 at 25 μM, the growth inhibitions were 50.1, 55.0 and 56.4% for the PC-3, K-565 and MCF-7 cell lines, respectively. This behavior was also observed when the concentration was diminished to 10 μM. Conjugate 8 showed higher activity than cisplatin against cell lines PC-3 and MFC-7. In the case of the conjugate 9, the antiproliferative activity was lower than with conjugate **8**. The free olsalazine and the free salicylic acid showed very low antiproliferative activity. None of the studied samples showed any activity against the control COS-7 cell line in comparison with free cisplatin (42.4%).

In general, the PC-3 (prostate grade IV adenocarcinoma), K-562 (chronic myelogenous leukemia) and MCF-7 (human mammary adenocarcinoma) cancer cell lines appeared to be more sensitive to growth inhibition by the olsalazine-PAMAM-dendrimer-salicylic acid-conjugates than the U-251, HCT-15 and SKLU-1 cancer cell lines. The compound 9 was the best inhibitor of this series against the PC-3, K-562 and MCF-7 cell lines with an IC_50_ of 17.5 ± 2.1 and 11.6 ± 3.6, respectively, and the activity was very close to that of cisplatin. Moreover, these compounds were tested against the COS-7 African green monkey kidney cell line (Table 2) in order to determine their selectivity for cancer cells compared to normal cells. All the compounds showed better selectivity for the cancer cells than for the COS-7 cell line.

## 4. Conclusions

In summary, we report the development of novel water-soluble olsalazine-PAMAM-dendrimer-salicylic acid-conjugates that inhibit the proliferation of the PC-3, K-562, HCT-15 and MCF-7 cell lines in vitro. The drug release was followed by UV spectroscopy at pH 7.4. These conjugates were designed and synthesized for maximal MCF-7 inhibition and exhibited high comparable selectivity against human mammary adenocarcinoma cancer cells. The olsalazine-PAMAM-dendrimer-salicylic acid-conjugate **9** revealed an IC_50_ value that was lower than that of cisplatin for the MCF-7 cell line. In addition, the use of olsalazine-PAMAM-dendrimer-salicylic acid-conjugates could be an effective way to diminish the damage towards normal cells.

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
