# Peer review of "Anticancer Activity of Water-Soluble Olsalazine-PAMAM-Dendrimer-Salicylic Acid-Conjugates"

_biomolecules, 2019, doi:10.3390/biom9080360_

Round 1
Reviewer 1 Report
The authors describe the synthesis of two generations of PAMAM dendrimers with diazo-salicylic acid at the core and salicylic acid conjugated to the surface. They then look at release of amino-salicylic acid and salicylic acid from these products in aqueous buffer and at the products cytotoxic activity against several cancer cell lines.
The introduction is satisfactory, although a quick keyword search found another salicylic acid/PAMAM dendrimer paper that should be referenced here: Nano Lett. 2016 April 13; 16(4): 2248–2253. doi:10.1021/acs.nanolett.5b04517. Polishing of English is required in the abstract and the supplementary information has a lot of spelling mistakes and poor English.
The vertical scales in Figures 2 and 3 are said to be in percent, but in reference to the text, the numbers should say 20, 40 etc. not 0.2, 0.4 etc. as they currently do to be truly percent numbers. The monitoring of release of 5-amino-salicylic acid and salicylic acid is done by UV at two different wavelengths. The authors should demonstrate by references or presentation of UV absorbance traces that there is no absorbance overlap between the two species or between them and the starting compounds (or if there is an overlap, that this is taken account of).
It should be made clear in Table 2 whether the IC50 concentrations are concentration of compounds 8, 9 etc. or whether this is the effective concentration of salicylic acid in these compounds.
The statement is made that “compound 9 was the best inhibitor of this series against PC-3 and MFC cell lines” however, for MFC-7, the error bars for the IC50 results for compound 8 and 9 overlap and so stating that one is better than the other should be done with caution.
The supplementary data is concerning. There is no chromatographic purification in the whole manuscript and given the nature of PAMAM synthesis, the data shown demonstrates the mediocre purity of materials that you would expect in that situation. Even in the small molecule 1, a significant impurity can be seen in the 1H NMR. Where mass spectral data is given, there should be a statement of the associated calculated mass and what formula that calculated mass represents. For compound 3, the MS quoted in the experimental text is 843 whereas in Figure 13 it is 849.
A lot of the NMR data is poorly processed and many of the 2D NMR spectra should be reduced in amplitude.
Author Response
REVIEWER 1
The authors describe the synthesis of two generations of PAMAM dendrimers with diazo-salicylic acid at the core and salicylic acid conjugated to the surface. They then look at release of amino-salicylic acid and salicylic acid from these products in aqueous buffer and at the products cytotoxic activity against several cancer cell lines.
QUESTION: The introduction is satisfactory, although a quick keyword search found another salicylic acid/PAMAM dendrimer paper that should be referenced here: Nano Lett. 2016 April 13; 16(4): 2248–2253. doi:10.1021/acs.nanolett.5b04517.
ANSWER: The reference Nano Lett. 2016 was added.
QUESTION: Polishing of English is required in the abstract and the supplementary information has a lot of spelling mistakes and poor English.
ANSWER: The English in the abstract and the Supplementary Material was improved. The English was improved throughout the entire manuscript as well.
QUESTION: The vertical scales in Figures 2 and 3 are said to be in percent, but in reference to the text, the numbers should say 20, 40 etc. not 0.2, 0.4 etc. as they currently do to be truly percent numbers.
ANSWER: Yes, this was my mistake, but now the figure 3 is in percent numbers.
QUESTION: The monitoring of release of 5-amino-salicylic acid and salicylic acid is done by UV at two different wavelengths. The authors should demonstrate by references or presentation of UV absorbance traces that there is no absorbance overlap between the two species or between them and the starting compounds (or if there is an overlap, that this is taken account of).
ANSWER: The referee is right. We were measuring at two different wavelengths. This was not correct and now we measured only at the maximum wavelength (at 300 nm) observed in the UV spectrum of the conjugates. We added the graphics of the UV spectra of the drug release for the compounds 8 and 9 (Figure 2). And the text was also changed.
The olsalazine-PAMAM-dendrimer-salicylic acid-conjugates 8 and 9 showed a relatively slow release in the first minutes, followed by a sustained release after 500 minutes. Salicylic acid was released from the corresponding first generation 8 and the second generation 9. Figure 2 shows the time course of UV spectral change from the conjugates. The amount of drug released from the conjugates was calculated from their UV absorption at 300 nm at pH 7.4. The amount of drug released was constant regardless of the generation number. This result could be related to the relative lower stability of the conjugates in the above pH.
And for the figure 3 the text also was changed.
Drug release from both conjugates 8 and 9 was calculated from their UV absorption at 300 nm as depicted in Figure 3. At physiological pH 7.4, the dendrimer conjugate 8 showed a less controlled release pattern compared to the dendrimer conjugate 9.
QUESTION: It should be made clear in Table 2 whether the IC50 concentrations are concentration of compounds 8, 9 etc. or whether this is the effective concentration of salicylic acid in these compounds.
ANSWER: The concentration of the conjugates 8 and 9 was diluted in 4 and 8 times, respectively, to have the anticancer activity for one molecule of salicylic acid and compare it to the free salicylic acid to have the same concentration.
QUESTION: The statement is made that “compound 9 was the best inhibitor of this series against PC-3 and MFC cell lines” however, for MFC-7, the error bars for the IC50 results for compound 8 and 9 overlap and so stating that one is better than the other should be done with caution.
ANSWER: The referee is right. It is difficult to make this conclusion. For this reason the phrase was changed for "The olsalazine-PAMAM-dendrimer-salicylic acid-conjugate 9 revealed an IC50 value which was lower than that of cisplatin for the MCF-7 cell line."
QUESTION: The supplementary data is concerning. There is no chromatographic purification in the whole manuscript and given the nature of PAMAM synthesis, the data shown demonstrates the mediocre purity of materials that you would expect in that situation.
ANSWER: The purity of all the compounds was confirmed by 1H, 13 NMR in one and two dimensions, FTIR and UV-vis spectroscopies, FAB+, Electro spray or MALDI-TOF spectrometry and elemental analysis. We have also added the IR spectra in the Supplementary Material.
QUESTION: Even in the small molecule 1, a significant impurity can be seen in the 1H NMR. Where mass spectral data is given, there should be a statement of the associated calculated mass and what formula that calculated mass represents. For compound 3, the MS quoted in the experimental text is 843 whereas in Figure 13 it is 849.
ANSWER: The mass of the compound 3 is 843. It was confirmed by FAB+ mass spectrometry and now the mass spectrum is correct.
QUESTION: A lot of the NMR data is poorly processed and many of the 2D NMR spectra should be reduced in amplitude.
ANSWER: All the NMR signals were well assigned and some of the 2D NMR data were reduced in amplitude.

Reviewer 2 Report
The manuscript “Anticancer Activity of Water-Soluble Olsalazine-PAMAM-Salicylic Acid Conjugates” was found promising with good experimental design, however, more work is required to reach the quality of publication.
1- Very little evidence was presented to confirm the structure of the intermediate and final products. 1H NMR alone or with 13C NMR are not sufficient. MS and purity assessment (chromatography for example) are required. More characterisations are needed for the final product in the term of how many SA molecules were covalently attached to the dendrimer?
2- Stability studies in different pHs are also required. (The extracellular pH of tumour tissues is often acidic).
3- Very limited discussion. Mainly just presented the results.
4- Very poor abstract- more specific abstract with highlighted findings is required.
5- English review is required, avoid long sentences and use coordinating conjugations appropriately.
Author Response
REVIEWER 2
The manuscript “Anticancer Activity of Water-Soluble Olsalazine-PAMAM-Salicylic Acid Conjugates” was found promising with good experimental design, however, more work is required to reach the quality of publication.
QUESTION: 1- Very little evidence was presented to confirm the structure of the intermediate and final products. 1H NMR alone or with 13C NMR are not sufficient. MS and purity assessment (chromatography for example) are required.
ANSWER: All the compounds were characterized by MS.
QUESTION: More characterisations are needed for the final product in the term of how many SA molecules were covalently attached to the dendrimer?.
ANSWER: The attachment of all the SA in the dendrimer was confirmed by MS.
QUESTION: 2- Stability studies in different pHs are also required. (The extracellular pH of tumour tissues is often acidic).
ANSWER: The stability of conjugates was studied in conditions of gastric fluid pH 1.2 and intestinal fluid pH 7.4 for 2h and 12 h, respectively, where a negligible amount of 5‑ASA was hydrolyzed. No degradation of conjugates at pH 1.2 and 7.4 showed that the conjugates were stable and remained intact at the gastric pH.
QUESTION: 3- Very limited discussion. Mainly just presented the results.
ANSWER: The discussion was improved.
QUESTION: 4- Very poor abstract- more specific abstract with highlighted findings is required.
ANSWER: The abstract was improved.
QUESTION: 5- English review is required, avoid long sentences and use coordinating conjugations appropriately.
ANSWER: The English was improved throughout the manuscript.

Round 2
Reviewer 2 Report
All issues have been addressed
Author Response
Suggestion
1. Two issues are important in design of novel anticancer drugs: potency against cancer cell lines and low toxicity against normal lines. Therefore low cytotxicity against COS-7 in comparison with cisplatin should be emphasized in Abstract.
Answer:
The abstract now reads:
"Improving the activity and selectivity profile of anticancer agents is to design drug carrier systems employing soluble macromolecules. Thus, olsalazine-PAMAM-dendrimer-salicylic acid-conjugates with different length dendritic arms showed a good stability of the chemical link between drug and spacer. The drug release was followed in vitro by UV studies. Evaluation of the cytotoxicity of the olsalazine-PAMAM-dendrimer-salicylic acid-conjugates employing a sulforhodamine B (SRB) assay in PC-3 (human prostatic adenocarcinoma) and MCF-7 (human mammary adenocarcinoma) cell lines, demonstrated that the conjugate 9 was more active as an antiproliferative agent than cisplatin and no cytotoxicity towards the African green monkey kidney fibroblast (COS-7) cell line was observed with any of the conjugates synthesized in the present work".
Suggestion
2. Table 2: please define what NC stands for.
Answer: The word NC was defined under the table 2 as "Non Cytotoxic".
